



Combined use of Mie-Raman and fluorescence lidar observations for improving aerosol
characterization: feasibility experiment
Igor Veselovskii[1], Qiaoyun Hu[2], Philippe Goloub[2], Thierry Podvin[2], Mikhail Korenskiy[1], Olivier
Pujol[2], Oleg Dubovik[2], Anton Lopatin[3]
*[1]General Physics Institute, Vavilova st., 38, Moscow, 119991, Russia.*
*[2]Univ. Lille, CNRS, UMR 8518 - LOA - Laboratoire d'Optique Atmosphérique, Lille F-59000,*
*France*
*[3]GRASP-SAS, Villeneuve d'Ascq, France*
**Abstract**
To study the feasibility of a fluorescence lidar for aerosol characterization, the fluorescence
channel is added to LILAS - multiwavelength Mie-Raman lidar of Lille University, France. A
part of fluorescence spectrum is selected by the interference filter of 44 nm bandwidth centered
at 466 nm. Such an approach has demonstrated high sensitivity, allowing to detect fluorescence
signal from weak aerosol layers (backscattering coefficient at 1064 nm is below 0.02 $Mm^{-1}sr^{-1}$)
up to a height of 5000 m. Simultaneous detection of nitrogen Raman and fluorescence
backscatters allows to quantify the fluorescence backscattering coefficient. Observations were
performed during November 2019 – February 2020 period. The fluorescence capacity (ratio of
fluorescence to elastic backscattering coefficients) varied in a wide range, being the highest for
the smoke and the lowest for the dust particles. The fluorescence capacity depends as well
strongly on the relative humidity, because the water uptake at the condition of high RH increases
the elastic backscattering, without significant modification of the fluorescence. Thus,
simultaneous measurements of Mie-Raman and fluorescence lidars open opportunity for the
study of the particle hygroscopic growth. The fluorescence technique can be used also for
monitoring the aerosol inside the cloud layers. The results presented demonstrate, that aerosol
and cloud particles can be mixed both externally and internally. When the cloud is formed at the
top or inside the aerosol layer (such scenario can be probably considered as internal mixing) we
observed significant (up to factor 5) increase of fluorescence backscattering. Among possible
mechanisms of such enhancement we can assume modification of the scattering phase function
of the particles embedded in the water microspheres and the lens effect due to the water shell
presence.

**1. Introduction**





The aerosol – cloud interaction is one of the key factors influencing the Earth radiation
balance and, for its realistic modeling, knowledge of aerosol properties both outside and within
the cloud layer are needed. The multiwavelength Mie-Raman and HSRL lidars, measuring
aerosol backscattering and extinction coefficients at multiple wavelengths, are widely used for
remote characterization of aerosol properties (Burton et al., 2012). However, although useful for
studying aerosol, the amount of information contained in these measurements remains limited
(Burton et al., 2016; Alexandrov and Mishchenko, 2017). In addition, such lidars are not able to
detect and characterize aerosol inside a cloud layer, because aerosol scattering is masked by the
strong cloud particles scattering. To improve the lidar capability for aerosol characterization,
additional channels, measuring the laser induced fluorescence, can be used. Moreover, the
fluorescence provides unique opportunity to detect (and characterize) aerosol particles within
cloud layer, at least near the cloud base, thus allowing to investigate the aerosol – cloud
coexistence.
Fluorescence spectroscopy is a highly sensitive technique, widely used for the in-situ
monitoring of atmospheric organic particles (Pan et al., 2007, 2015; Miyakawa et al., 2015;
Huffman et al., 2019). The synergy of fluorimetry and lidar technology provides an opportunity
to perform such monitoring remotely (Immler et al., 2005; Rao et al., 2018; Saito et al., 2018).
Numerous types of atmospheric aerosols, such as biological particles, biomass burning and even
dust particles are fluorescent, being excited by UV radiation. When the excitation wavelength is
355 nm, the main part of emission spectra is usually contained within the 400–650 nm range
(Pan et al., 2015). The fluorescence spectrum varies with the aerosol types/composition, making
therefore possible their identification.
The recent interest in fluorescence lidars was stimulated also by the progress in the
development of the multianode photomultipliers allowing, in combination with spectrometer,
simultaneous detection of lidar signal in 32 spectral bins (Sugimoto et al., 2012; Reichardt et al.,
2014, 2017; Saito et al., 2018). Such multichannel detection has the obvious advantage to
analyze the whole spectrum, allowing the aerosol identification. However, sensitivity of such
lidar spectrometers is low when compared to the standard technique based on selection of
fluorescence spectrum intervals with interference filters (Immler et al, 2005; Rao et al., 2018; Li
et al., 2019). Such an approach, in addition to being more sensitive, allows more affordable
modification of a multiwavelength Mie-Raman lidar by adding one or more fluorescence
channels.
To obtain the highest sensitivity, it is mandatory to acquire the fluorescence in a wide
spectral range which, however, makes the data analysis more complicated, because variation of
aerosol and molecular transmission within the detection spectral range has to be accounted for.





In addition, in Mie-Raman multi-wavelength lidars one should avoid the spectral intervals
affected by elastic scattering and corresponding strong Raman lines. In this work, we present the
results of a feasibility experiment and evaluate the sensitivity of a single-channel fluorescence
lidar to different aerosol types. We focus in this study also on the possibility to monitor the
aerosol fluorescence within a cloud layer. However, to obtain quantitative measurements of
aerosol content within a cloud layer, several factors should be considered.

First of all, external/internal mixtures of aerosol particles with water droplet should be

distinguished. In case of external mixture, the fluorescence signal can be used directly for the
estimation of the particle concentration, if the particle type and corresponding fluorescence cross
section are known. For internal mixture (aerosol particle is inside the water droplet) the
fluorescence backscattering can be enhanced due to modification of the scattering phase function
of the particles embedded in the water microspheres (Kerker and Druger, 1979; Veselovskii et
al., 2002a). For insoluble particle, the presence of the water shell can lead also to an additional
increase of the fluorescence, due to the water droplet lens effect. Similar effect is well known for
the soot particles covered by non-absorbing shell (Schnaiter, 2005).

In our paper we present the results of fluorescence measurements performed at

*Laboratoire d'Optique Atmosphérique* (LOA) during November 2019 – February 2020 period.
During that period, the aerosol load was very low, so we were not able to determine the particle
properties from multiwavelength observations. The objective was then to estimate the efficiency
/added value of the fluorescence channel. We therefore mainly focus on analysis of efficiency of
fluorescence lidar monitoring of different types of aerosol and on detection of aerosol particles
inside low level cloud layer.

**2. Experimental setup and data analysis**

The measurements were performed using the LILAS - multiwavelength Mie-Raman

lidar, based on a tripled Nd:YAG laser with a 20 Hz repetition rate and pulse energy of 70 mJ at
355 nm. The backscattered light is collected by a 40 cm aperture Newtonian telescope. The
system is designed for simultaneous detection of elastic and Raman backscatters, allowing the so
called $3\beta+2\alpha+3\delta$ data configuration, including three particle backscattering ($\beta$), two extinction
($\alpha$) coefficients along with three depolarization ratios ($\delta$). Description of the system can be found
in recent publication of Hu et al., 2019. The aerosol extinction and backscattering coefficients at
355 and 532 nm were calculated from Mie-Raman observations (Ansmann et al., 1992) while
$\beta_{1064}$ was derived by the Klett method (Klett, 1985).

For the experiment described, the system was modified: water vapor 408 nm Raman filter

was replaced by a fluorescence one. Corresponding optical scheme together with transmission





curve of the interference filter in the fluorescence channel are shown in Fig.1. The nitrogen
Raman and fluorescence optical signals are separated by a dichroic mirror: more than 98% of
387 nm radiation is reflected and more than 95% of fluorescence signal is transmitted. For both
nitrogen Raman and fluorescence channels, the R9880U-01 PMTs were used. A part of the
wideband fluorescence signal was selected by an Alluxa interference filter centered at 466 nm
with 44 nm bandwidth. The filter transmission, at maximum, exceeds 98%. The operational band
was chosen outside of the overtones of $O_2$ and $N_2$ vibrational Raman lines. In addition, the
transmission of the selected fluorescence filter band matches the maxima of fluorescence of
many organic molecules (Saito et al., 2018; Reichardt et al., 2017). Filter provides OD6
suppression outside the transmission band. To increase the suppression, two identical
interference filters were used in tandem. For additional rejection of elastic scattering at 355 nm
and 532 nm the two-band notch filter was used. With such design, we estimate that the total
suppression of elastic scattering in the fluorescence channel is above OD14. In this paper,
observations were carried out during night-time only.

In an elastic channel, the backscattered radiative power $P_L$, at distance $z$ is described by

the lidar equation
$$P_L = O(z)\frac{1}{z^2}C_L(\beta_L^a + \beta_L^m)\exp\left\{-2\int_0^z (\alpha_L^a + \alpha_L^m)dz'\right\} = O(z)\frac{1}{z^2}C_L(\beta_L^a + \beta_L^m)T_L^2 \qquad (1)$$
Here $O(z)$ is the geometrical overlap factor, which is assumed to be the same for elastic, Raman
and fluorescence channels. $C_L$ is the range independent constant, including efficiency of
detection channel. $T_L$ is one-way transmission, describing light losses on the way from the lidar
to distance $z$ at laser wavelength $\lambda_L$. Backscattering and extinction coefficients contain aerosol
and molecular contributions: $\beta_L^a + \beta_L^m$ and $\alpha_L^a + \alpha_L^m$, where the superscripts "a" and "m" indicate
aerosol and molecular scattering, respectively.

In a Raman channel, the backscatter radiative power, $P_R$, can be rewritten as:

$$P_R = O(z)\frac{1}{z^2}C_R\beta_R\exp\left\{-\int_0^z (\alpha_L^a + \alpha_R^a + \alpha_L^m + \alpha_R^m)dz'\right\} = O(z)\frac{1}{z^2}C_R\beta_R T_L T_R \qquad (2)$$
Here $T_R$ is the atmospheric transmission at Raman wavelength $\lambda_R$. Raman backscattering
coefficient is:
$$\beta_R = N_R\sigma_R, \qquad (3)$$
where $N_R$ is the number of Raman scatters (per unit of volume) and $\sigma_R$ is the Raman differential
scattering cross section in the backward direction. To account for spectral dependence of aerosol
extinction, the Angstrom exponent $\gamma$ is used:


$$\frac{\alpha_L^a}{\alpha_R^a} = \left(\frac{\lambda_R}{\lambda_L}\right)^{\gamma}$$  (4)
The aerosol backscattering and extinction coefficients can be computed from Mie – Raman lidar
observations using equations (1-4), as shown by Ansmann et al. (1992).
In the case of the fluorescence, the emitted wavelengths are spread over wide spectral
range, so the spectral dependence of aerosol and molecular extinction coefficients, inside the
fluorescence band, should be considered. Moreover, the spectral differential fluorescence cross
section $\frac{d\sigma_F}{d\lambda}(\lambda, r)$ depends on particle size (Hill, et al., 2015), so the particle number size
distribution $\frac{dN(r)}{dr}$, which is the number of particles with radii between $r$ and $r+dr$ per unit of
volume, has to be considered. The radiative power in the fluorescence channel within the
spectral interval [$\lambda_{min},\lambda_{max}$] is:
$$P_F = O(z)\frac{1}{z^2}T_L \int_{\lambda_{min}}^{\lambda_{max}} \int_{r_{min}}^{r_{max}} C_F(\lambda) \times \frac{dN(r)}{dr} \times \frac{d\sigma_F}{d\lambda}(\lambda, r) \times \exp\left\{-\int_0^z [\alpha^a(\lambda, z') + \alpha^m(\lambda, z')]dz'\right\}drd\lambda$$  (5)
The spectral dependence of $C_F(\lambda)$ is determined mainly by the transmission of the interference
filter in the fluorescence channel. If the filter spectral width $\lambda_{max} - \lambda_{min}$ is not very high, the
procedure of data analysis can be simplified. The atmospheric transmission for fluorescence
signal
$$T_F(\lambda) = \exp\left\{-\int_0^z [\alpha^a(\lambda, z') + \alpha^m(\lambda, z')]dz'\right\}$$  (6)
can be taken at wavelength $\lambda_F$, corresponding to the center of the filter transmission band
$T_F(\lambda) = T_F(\lambda_F) \equiv T_F$. The filter transmission used (Fig.1) is close to rectangular and sensitivity
of the PMT used doesn't vary significantly within [$\lambda_{min}$, $\lambda_{max}$] interval, which means the
calibration constant $C_F$ can be considered as spectrally independent. Expression (5) can be
rewritten, by introducing the fluorescence backscattering coefficient $\beta_F$:
$$\int_{\lambda_{min}}^{\lambda_{max}} \int_{r_{min}}^{r_{max}} \frac{dN(r)}{dr} \times \frac{d\sigma_F}{d\lambda}(\lambda, r)drd\lambda = \int_{r_{min}}^{r_{max}} \frac{dN(r)}{dr} \times \sigma_F(r)dr = \beta_F$$  (7)
Here $\sigma_F(r) = \int_{\lambda_{min}}^{\lambda_{max}} \frac{d\sigma_F}{d\lambda}(\lambda, r)d\lambda$ is the effective fluorescence differential cross section, integrated
over spectral interval [$\lambda_{min}, \lambda_{max}$]. The use of $\beta_F$ allows to rewrite equation (5) for the power of
the fluorescence backscattering similarly to the Raman one.





$$P_F = O(z)\frac{1}{z^2}C_F\beta_F T_F T_L \qquad (8)$$
The fluorescence backscattering coefficient, $\beta_F$, can be obtained from the ratio of
equations (8) and (2) for fluorescence and Raman backscatters:
$$\beta_F = \frac{C_R}{C_F}\frac{P_F}{P_R}N_R\sigma_R\frac{T_R}{T_F} \qquad (9)$$
The ratio of atmospheric transmissions at $\lambda_R$ and $\lambda_F$ wavelengths ($T_R/T_F$) can be calculated the
same way as for water vapor measurements (Ansmann et al., 1992; Whiteman et al., 2006). In
our study, for the nitrogen molecule we used Raman scattering cross section at 355 nm
$\sigma_R=2.1*10^{-30}$ cm$^2$ (Burris et al., 1992), but, to obtain absolute values of $\beta_F$, $C_R/C_F$ ratio must be
determined. This ratio can be found from calibration, performed by using a lamp with known
spectrum, as it has been done for the Raman water vapor lidars (Venable et al., 2011). At current
stage, for $\beta_F$ estimations, we assume that sensitivities of PMTs in both channels are similar
(switching of PMTs between $N_2$-Raman and fluorescence channels didn't change results
noticeably), thus only difference in transmission of interference filters was considered. In all
results presented below $C_R/C_F=0.7$ value was used for calculations.
To characterize the efficiency of the fluorescence respect to elastic scattering, it is
convenient to consider also the particle fluorescence capacity, $G_F = \dfrac{\beta_F}{\beta_L}$, which is the ratio of
fluorescence and aerosol elastic backscattering coefficients (Reichardt et al., 2017). Here and
below, for simplicity, we will use notation $\beta^a \equiv \beta$. The aerosol loading in the atmosphere during
the experiment was very low and, in order to decrease the interference of the Raleigh scattering,
the backscatter at 1064 nm was mainly used for aerosol characterization, while for the cloud
layers the backscattering coefficients at 355 and 532 nm were used as well.
Multiwavelengtn Mie-Raman lidar measurements allow estimation of the particle number
density $\quad N = \displaystyle\int_{r_{\min}}^{r_{\max}}\frac{dN(r)}{dr}dr\quad$ as well as their total volume V (Müller et al., 1999; Veselovskii et
al., 2002b), thus a mean fluorescence cross section per a single particle can be estimated as
$\sigma_F^N = \dfrac{\beta_F}{N}$. Assuming, that in the simplest case, a fluorescence backscattering coefficient is
proportional to the particle volume, we can estimate the fluorescence cross section per a unit
particle volume as $\sigma_F^V = \dfrac{\beta_F}{V}$. Thus, synergy of Mie-Raman and fluorescence lidar measurements
should allow remote characterization of the particle fluorescent properties.





## 3. Observation results.

### *3.1. Fluorescence of aerosol layers.*

The measurements reported were performed during November 2019 – February 2020 period at the Lille Atmospheric Observation Platform (https://www-loa.univ-lille1.fr/observations/plateformes.html?p=apropos) hosted by Laboratoire d'Optique Atmospherique, University of Lille, Hauts-de-France region. Two examples of measurement are presented in Fig.2 and are showing height–temporal distributions of the range corrected lidar signal (RCS) at 1064 nm, of volume depolarization ratio, $\delta_{1064}$, and of fluorescence backscattering coefficient, $\beta_F$, for the nights 29-30th November 2019 and 6-7th February 2020.

During the first night (left column in Fig. 2), aerosol layer is localized mainly below 2000 m. Though the aerosol loading is low ($\beta_{1064}<0.01$ Mm$^{-1}$sr$^{-1}$) above 2000 m, it is well revealed by the enhanced depolarization ratio and the enhanced fluorescence backscattering coefficient. During the second night of observation (right column in Fig.2), a detached/isolated layer is observed at approximately 3000 m. This layer is characterized by high depolarization ratio (the particle depolarization ratio at 1064 nm in the center of the layer exceeds 15%), indicating to the presence of dust. An explanation of the observed increase of fluorescence signal could be mixing of mineral dust particles with organic materials (Sugimoto et al., 2012; Miyakawa et al., 2015) and local aerosol during transportation.

The time averaged profiles ($\beta_{1064}$, $\beta_F$, $G_F$) for these two nights, as well as for 16th January episode are shown in Fig.3. Backscattering coefficient $\beta_{1064}$ was calculated by Klett method, assuming a lidar ratio S=50 sr. Due to low aerosol extinction value, the results are not sensitive to the choice of S. The closest available radiosonde data are from the Herstmonceux (UK) and Essen (Belgium) stations, located 160 km and 120 km away from the observation site respectively. Data from both stations show that on the night 29-30 November 2019 the relative humidity (RH) was about 70% at 1000 m and dropped below 20% above 2000 m. Pure water is not fluorescing, so the water uptake by the particle, in the condition of high relative humidity (RH), is expected to yield an increase of elastic scattering without significant effect on the fluorescence emission. The aerosol backscattering $\beta_{1064}$ on 29-30th November (Fig.3a) is 0.4 Mm$^{-1}$sr$^{-1}$ at 1000 m and decreases by a factor 40 at 1900 m, while $\beta_F$ within this height range changes less than twice. This is supporting the assumption that the observed variation of aerosol backscattering in the PBL is mainly due to the change of the particle water fraction. The water uptake at low altitudes agrees with low values of the observed particle depolarization ratio $\delta_{1064}^p$, which is below 0.5% at 1000 m. Within weak aerosol layer at the range 2500 – 4000 m, the particle depolarization $\delta_{1064}^p$ is about 5% and we observe the increase of fluorescence capacity $G_F$,



224 with respect to the layer below 2000 m, up to $2.5*10^{-4}$. This increase of $G_F$ in the 2500 – 4000 m

225 layer can be due to the presence of another particles type, for example, biomass burning. From

226 this episode, one can conclude that fluorescence backscattering, though being almost 4 orders

227 lower than elastic one, can be reliably detected with our current lidar configuration.

228  On January $16^{th}$ (Fig.3b), atmospheric RH also decreases with height, from about 80% at

229 1000 m to less than 20% above 2000 m, leading to an increase of $G_F$ for more than one order of

230 magnitude. Such variation of $G_F$ within the PBL is probably also related to the particle water

231 uptake, just like in Fig.3a. Aerosol backscattering increases above 3000 m and reaches its

232 maximum value at 5000 m. Within 3000 m – 5500 m range, fluorescence capacity was about

233 $2.5*10^{-4}$, which is higher than in the PBL.

234  On February $6\text{-}7^{th}$ the aerosol loading in the PBL is very low ($\beta_{1064}<0.003$ Mm$^{-1}$sr$^{-1}$ at 1000

235 m) and RH from radiosonde at Herstmonceux is below 40% in the height range considered. At

236 3000 m, a dust layer is observed (Fig.3c). In the middle of this layer, fluorescence capacity is

237 about $0.6*10^{-4}$ which is about factor 4 lower than in the elevated layers in Fig.3a,b. Still,

238 significant value of $G_F$ can indicate the presence of organic materials in the dust layer (Sugimoto

239 et al., 2012).

240  As discussed in section 2, lidar measurements provide an opportunity to estimate the

241 particle fluorescence cross section. For this, we need to know the particle number N and volume

242 V density in the aerosol layer, which, in principle, can be determined from the multiwavelength

243 lidar observations (Muller et al., 1999; Veselovskii et al., 2002b). In our case, however, due to

244 very low aerosol loading the extinction coefficients could not be determined. Still, the rough

245 estimations of the particle parameters can be done using the predefined aerosol model driven by

246 only a few parameters. In our study we use a simplified approach, modeling aerosol as an

247 external mixture of several aerosol components with predetermined properties. The definition of

248 aerosol components is based on global multiyear AERONET observations (Dubovik et al., 2002)

249 with some modifications. All aerosol types are described by a bimodal particle size distribution

250 (PSD)

$$\frac{dV}{d\ln r} = \sum_{i=f,c} \frac{C_{V,i}}{\sqrt{2\pi}\sigma_i} \exp\left[ -\frac{(\ln r - \ln r_{V,i})^2}{2\sigma_i^2} \right]$$
(10)

252 where $C_{V,i}$ denotes the particle volume concentration, $r_{V,i}$ is the median radius, and $\sigma_i$ is the

253 standard deviation. Subscripts $f$ and $c$ correspond to the fine and coarse mode respectively. The

254 parameters of the number size distribution $\dfrac{dN}{d\ln r}$ can be obtained from (10) using the

255 expressions from Horvath et al. (1990). Table 1 shows the model parameters for three aerosol



types: biomass burning (BB), urban (UR) and dust (DU). From this model, the aerosol
backscattering and extinction coefficients can be calculated at any wavelength. As mentioned
above, due to low aerosol loading, we use only backscattering coefficient at 1064 nm, so Table 1
presents $\beta_{1064}^{N} = \dfrac{\beta_{1064}}{\int_{r_{\min}}^{r_{\max}} \dfrac{dN(r)}{dr} dr}$ - mean backscattering coefficient for a single particle (N=1),
together with corresponding complex refractive index (CRI) used in computations. Calculations
were performed in assumptions of spherical particles for BB and UR and for the randomly
oriented spheroids for dust (Dubovik, et al., 2006). The volume $V^{N=1}$ in the Table 1 is also given
for N=1 (so can be considered as a single particle average volume). Thus, if the aerosol type is
known, comparing of computed $\beta_{1064}^{N=1}$ from Table 1 with observed values $\beta_{1064}$, yields the
number and volume particle densities as $N = \dfrac{\beta_{1064}}{\beta_{1064}^{N=1}}$ and $V = N \times V^{N=1}$.
Table 2 summarizes for the three nights from Fig.3, the fluorescence cross sections per a
single particle, $\sigma_F^N = \dfrac{\beta_F}{N}$, and per unit volume, $\sigma_F^V = \dfrac{\beta_F}{V}$. Values are provided for the altitudes
corresponding to the maximum of fluorescence backscattering $\beta_F$ in elevated layers, where the
relative humidity (RH) should be low and hygroscopic effect reduced. Particles are assumed to
be of biomass burning origin for November 30th and January 16th, and from dust origin for
February 6-7th. We should remind, however, that our estimations of N (and so $\sigma_F^N$) are rough,
since they depend on the assumed aerosol type. The particle volume, V, is however a more
reliable parameter. For example, if the UR aerosol type is considered, rather than the BB one, the
particle number density, N, for November 30th becomes N=21cm$^{-3}$ (instead 63 cm$^{-3}$ for BB)
while the total volume remains rather constant (V=0.34 μm$^3$·cm$^{-3}$ instead of 0.37 μm$^3$cm$^{-3}$).
Thus, presentation of cross section per a unit of volume $\sigma_F^V$ appears more trustable. The
fluorescence cross sections $\sigma_F^V$ for November 30th and January 16th are very close, but for the
dust layer (February 6-7th, 2020), the cross section is about a factor 4 lower. From the data
presented it is also possible to estimate the spectral differential cross section, $\dfrac{\sigma_F^V}{\Delta\lambda}$, where $\Delta\lambda$ is
the width of the filter transmission band.
It is rather difficult to validate our values of the fluorescence differential cross section. We
nevertheless compare them to in situ ground-based fluorescence measurements. Such reference
data are available mainly for biological particles (e.g. Pan, 2015). For biological particles, the





highest $\dfrac{d\sigma_F}{d\lambda}$ value, for a single particle with diameter 1.2 μm – 3.0 μm varies in the range (1-
100)*$10^{-15}$ $cm^2sr^{-1}nm^{-1}$ when stimulating radiation at 365 nm is used (Pan, 2015). Thus, our
estimated values look reasonable, keeping in mind that the fluorescence cross section of the
biological particles is higher than that of smoke.

*3.2. Fluorescence of aerosol particles within cloud layers*

One of the attractive features of the fluorescence technique is the possibility to detect

aerosol and derive its content within the cloud layer. However, aerosol and cloud particles can be
mixed externally or internally. In the first case, the estimation of the dry particle volume is more
or less straightforward, using the fluorescence cross sections obtained from the measurements in
the aerosol layers. For the internal mixing, aerosols are located inside the water droplet in solid
or dissolved state. It is known, that the fluorescence of microspheres can be increased in the
backward direction by factor ~2, comparing to fluorescence of a bulk material, due to
modification of the scattering phase function (Kerker et al., 1979; Veselovskii et al., 2002a).
Moreover, a fluorescence cross section of solid substance may differ from dissolved one. All this
complicates the quantification of the measurements for the case of internal mixing.

The results from November 13th and 18th, representing an **external** mixture case of

aerosol and clouds particles, are shown in Fig. 4. The backscattering coefficients are given at 532
nm, because in the cloud layers the detector in 1064 nm channel was sometimes saturated. On
November 13th, we observe a strong oscillation of $\beta_{532}$, due to the contribution of the cloud
layers within 1000 m – 3000 m. The increase of $\beta_{532}$, however, does not lead to significant
enhancement of the fluorescence backscattering $\beta_F$ in the range of 1000—3000 m.

On 18th November, the cloud layer within 1500 – 2000 m range exhibit an even stronger

elastic backscattering, exceeding 80 $Mm^{-1}sr^{-1}$. However, no significant change of fluorescence
backscattering is observed. These results clearly indicate the absence of leaks/contamination of
elastic scattering in the fluorescence channel, which in turn allow us monitor fluorescence within
cloud layers.

The situation, however, can be different, when the cloud droplets are formed on the aerosol

particles, thus fluorescent aerosols are inside the water particles. Fig.5 shows the height –
temporal distributions of the lidar signal at 1064 nm and the fluorescence backscattering
coefficient on the night 19-20th November 2019.

After 21:50 UTC a thin cloud layer starts to form at the top of the PBL resulting in

simultaneous increase of $\beta_F$. To quantify the influence of cloud water droplet on the fluorescence
backscattering, Fig.6a provides profiles of aerosol and fluorescence backscattering coefficients



for two temporal intervals 20:00 – 21:30 UTC and 21:30 – 00:30 UTC, prior and after the cloud
layer formation respectively. Prior to cloud formation the aerosol load is very low, so
backscattering is provided only at 1064 nm and to be distinguished at this figure, the value of
$\beta_{1064}$ is multiplied by factor 100. For 20:00 – 21:30 $\beta_{1064}$ at 1500 m (height where the cloud
forms) is about 0.07 Mm$^{-1}$sr$^{-1}$. After the cloud formation the backscattering coefficient is shown
at 532 nm, because 1064 nm detector in the cloud layer was overloaded. $\beta_{532}$ increases up to 500
Mm$^{-1}$sr$^{-1}$ (maximum value) and the fluorescence backscattering increases by approximately
factor 5.
Similar scenario occurred on the night 23-24th November 2019 (Fig.6b). Prior to the cloud
formation (21:00 – 23:00 UTC) the backscattering coefficient at 900 m height is $\beta_{1064}$=0.02 Mm$^{-}$
$^{1}$sr$^{-1}$ and after cloud formation $\beta_{532}$ increases up to 130 Mm$^{-1}$sr$^{-1}$, while $\beta_F$ again increases about
factor 5. The profiles of $\beta_F$ in Fig.6 prior and after the cloud formations remain the same, below
the cloud. It corroborates the suggestion that the cloud was not transported by the air masses with
different properties, but the process of water vapor condensation occur, leading to formation of
internal mixture of aerosol and cloud particles.
We should recall also, that the enhancement of $\beta_F$ can not be explained by just insufficient
suppression of elastic scattering. The enhancement was observed only inside an aerosol layer,
while clouds with similar backscattering coefficients, but outside the aerosol layer, didn't
provide the increase of $\beta_{532}$. Several possible mechanisms could be responsible for the observed
fluorescence backscattering increase. As mentioned above, the fluorescence scattering phase
function of microspheres can have a peak in the backward direction (Pan et al., 2002;
Veselovskii et al., 2002a). The fluorescence cross section of particles may be subject to
hygroscopic effect so impacted by the amount of available water vapor (RH). For example, it is
known, that fluorescence cross section of wet bacterial spores is higher than that of dry ones
(Kunnil et al., 2004). For insoluble particles, the presence of a water shell can lead also to
additional increase of the fluorescence, due to the lens effect produced by the droplet. Similar
effect is well known for the soot particles covered by non-absorbing shell (Schnaiter et al.,

2005).

One of the objectives of this study was to discuss the possibility to derive the aerosol
content within cloud layers, from the fluorescence measurements. Fig.7 shows, for November
15$^{th}$ 2019, 2:45 – 6:15 UTC, the height – temporal distributions of the range corrected lidar
signal at 1064 nm and the fluorescence backscattering $\beta_F$. Low cloud layers appear at
approximately 2000 m and a signal of aerosol fluorescence is observed within this layer up to
3000 m. The vertical profiles of $\beta_{532}$ and fluorescence backscattering $\beta_F$, integrated over this
temporal interval, are shown in Fig.8a. Fluorescence backscattering does not change



significantly inside 1000 – 3500 m height range, thus aerosol content outside and inside the
cloud layer is similar. Strong increase of elastic scattering at 2700 m does not lead to significant
increase of $\beta_F$, hence the mixture of aerosol and cloud particles can be considered as external.
Using the results from the previous section for UR aerosol model, the volume density of the
ambient particles within the cloud layer can be estimated as ~2 $\mu m^3 cm^{-3}$.
On November 25th 2019 (Fig.8b), a low cloud layer at 850 m leads to increase of $\beta_F$ by
approximately of factor 2, in a similar way as in Fig.6. However, in the cloud above 2000 m, no
fluorescence signal, exceeding the noise level, is observed, indicating that the cloud is free of
aerosol. Thus, the results in Fig.8 demonstrate, that the fluorescence technique is capable to
distinguish between "clean of particles" and "polluted" cloud layers.

**Conclusion**
In our research we analyzed the feasibility of the fluorescence channel, added to the
multiwavelength Mie-Raman lidar for aerosol characterization. The results obtained,
demonstrate that the use of an interference filter for selection the part of the fluorescence
spectrum allows highly efficient lidar operation. In particular, LILAS lidar with the interference
filter of 44 nm width in the fluorescence channel, was able to detect fluorescence signal from
weak aerosol layers ($\beta_{1064}$< 0.02 $Mm^{-1}sr^{-1}$) up to 5000 m. During the experiment the fluorescence
capacity $G_F = \dfrac{\beta_F}{\beta_{1064}}$ of aerosol at condition of low RH varied through the $(0.6 – 2.5)\times10^{-4}$ range,
being the highest for the smoke and the lowest for the dust particles. The fluorescence capacity
depends as well on the relative humidity, because the water uptake at the condition of high RH
increases the elastic backscattering, without significant modification of the fluorescence. Thus
simultaneous measurements of Mie-Raman and fluorescence lidar open opportunity for the study
of the particle hygroscopic growth.
The lidar measurements, in principle, allow to get the quantitative information about the
aerosol fluorescence cross section in the elevated aerosol layers. For several atmospheric
situations the rough estimations of $\sigma_F$ were performed in this study and the results obtained look
reasonable, comparing with published values for biological particles. Still these results should be
taken as preliminary and the next important step in quantification of the fluorescence
measurements will be the system calibration, using a lamp with known spectrum. As well, more
deep comparison of $\sigma_F$ obtained from the lidar and laboratory measurements, for different
aerosol types, is needed for validation. It should be mentioned, that the fluorescence and
multiwavelength Mie-Raman lidar techniques are complimentary. The multiwavelength lidar



allows aerosol typing and estimation of the particle number and volume densities, that are later
used to derive the fluorescence cross sections from observed $\beta_F$. The fluorescence
measurements, in turn, help to improve the aerosol classification. The synergy of fluorescence
and multi-wavelength lidar techniques was not realized in this study, due to too low aerosol
loading in November – February period. However, we plan new experiments during Spring –
Summer season, when AOD is larger in Lille.

Results presented demonstrate also, that the fluorescence technique can be efficiently

used to estimate the particle concentration inside the cloud layer (at least near the cloud base, if
penetration depth of the laser radiation is small), which is important in the study of aerosol –
cloud interaction. Moreover, our measurements indicate, that aerosol and cloud particles can be
mixed both externally and internally. In the clouds formed at the top or inside the aerosol layer
(which makes possible internal mixture scenario), significant increase of the fluorescence
backscattering coefficient (up to factor 5), was observed. Among possible mechanisms of such
enhancement we can assume the increase of fluorescence cross section in solvent state,
modification of the scattering phase function of particles embedded in microspheres and the lens
effect due to the water shell. At a moment we are not able to specify the mechanism and future
studies are needed.

In coming studies we plan additional modifications of the lidar. The efficiency of the

fluorescence channel is demonstrated to be high, so the filter bandwidth can be decreased to 10-
20 nm, which will make more solid our simplified approach to data analysis. We consider also
the possibility to add second fluorescence channel near 550 nm, which should improve
selectivity of the fluorescence technique to different aerosol types. The water vapor channel will
be returned back to the system, which is essential for the study of particle hygroscopic growth.
Collocated measurements of the microwave radiometer of the Laboratoire d'optique
atmosphérique will be used to derive the RH profiles. When considering, the aerosol
characterization inside the cloud, the possible effect of the multiple scattering on the results will
be also analyzed.

***Acknowledgments***
The authors are very grateful to Service National d'Observation PHOTONS/AERONET-
EARLINET from CNRS-INSU, France; to ACTRIS-2 program under the European Union's
Horizon 2020 research and innovation programme under grant agreement no. 654109; to Région
Hauts-de-France and the Ministère de l'Enseignement Supérieur et de la Recherche (CPER
Climibio); and to IDEAS+/ESA Programme for supporting this research.



Table 1. Parameters of the biomass burning (BB), URban and DUst particles used in the model.
The volume $V^{N=1}$ and backscattering coefficient $\beta_{1064}^{N=1}$ are given for a single particle (N=1).

| Type | $r_{V,f}$ μm | $r_{V,c}$ μm | $\sigma_f$ | $\sigma_c$ | $\dfrac{C_{V,f}}{C_{V,c}}$ | CRI$_{1064}$ | $V^{N=1}$ μm$^3$/cm$^3$ | $\beta_{1064}^{N=1}$ Mm$^{-1}$sr$^{-1}$ |
|---|---|---|---|---|---|---|---|---|
| BB | 0.12 | 3.95 | 0.4 | 0.75 | 1.32 | 1.51-i.0.02 | 5.91E-3 | 1.58E-4 |
| URban | 0.175 | 3.275 | 0.38 | 0.75 | 2.5 | 1.4-i0.003 | 1.61E-2 | 4.69E-4 |
| DUst | 0.12 | 2.32 | 0.4 | 0.6 | 0.05 | 1.56- i0.001 | 7.6E-2 | 2.83E-3 |

Table.2. The aerosol parameters in elevated layers for three measurement sessions from Fig.3,
including the fluorescence $\beta_F$ and aerosol $\beta_{1064}$ backscattering coefficients, number N and
volume V particle densities, the differential fluorescence cross sections per a single particle $\dfrac{\sigma_F}{N}$
and per unit volume $\dfrac{\sigma_F}{V}$, together with spectral density $\dfrac{\sigma_F}{V\Delta\lambda}$.

| Date | Height km | $\beta_F$, Mm$^{-1}$sr$^{-1}$ | $\beta_{1064}$, Mm$^{-1}$sr$^{-1}$ | N, cm$^{-3}$ | V, μm$^3$cm$^{-3}$ | $\dfrac{\sigma_F}{N}$, 10$^{-15}$ cm$^2$sr$^{-1}$ | $\dfrac{\sigma_F}{V}$, 10$^{-13}$ cm$^2$sr$^{-1}$μm$^{-3}$ | $\dfrac{\sigma_F}{V\Delta\lambda}$, 10$^{-15}$ cm$^2$sr$^{-1}$μm$^{-3}$nm$^{-1}$ |
|---|---|---|---|---|---|---|---|---|
| 30 Nov | 4.0 | 3.0E-6 | 0.010 | 63 | 0.37 | 0.48 | 0.81 | 1.84 |
| 16 Jan | 5.0 | 4.88E-6 | 0.013 | 82 | 0.60 | 0.48 | 0.81 | 1.84 |
| 6-7 Feb | 2.9 | 5.63E-6 | 0.096 | 34 | 2.58 | 2.18 | 0.22 | 0.5 |






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


**Figures**




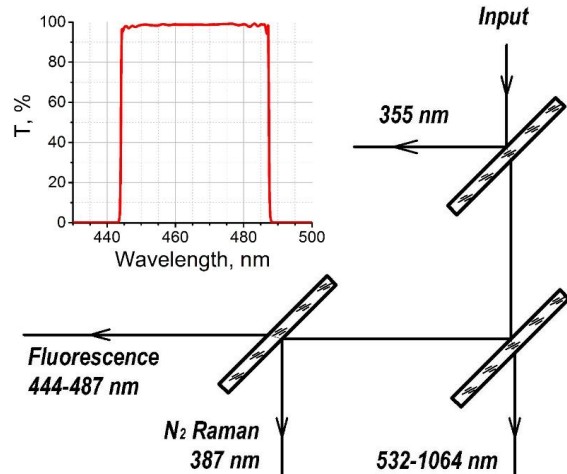


Fig.1 Optical scheme of the elastic, Raman and fluorescence backscatters separation together
with transmission curve of the interference filter in the fluorescence channel.









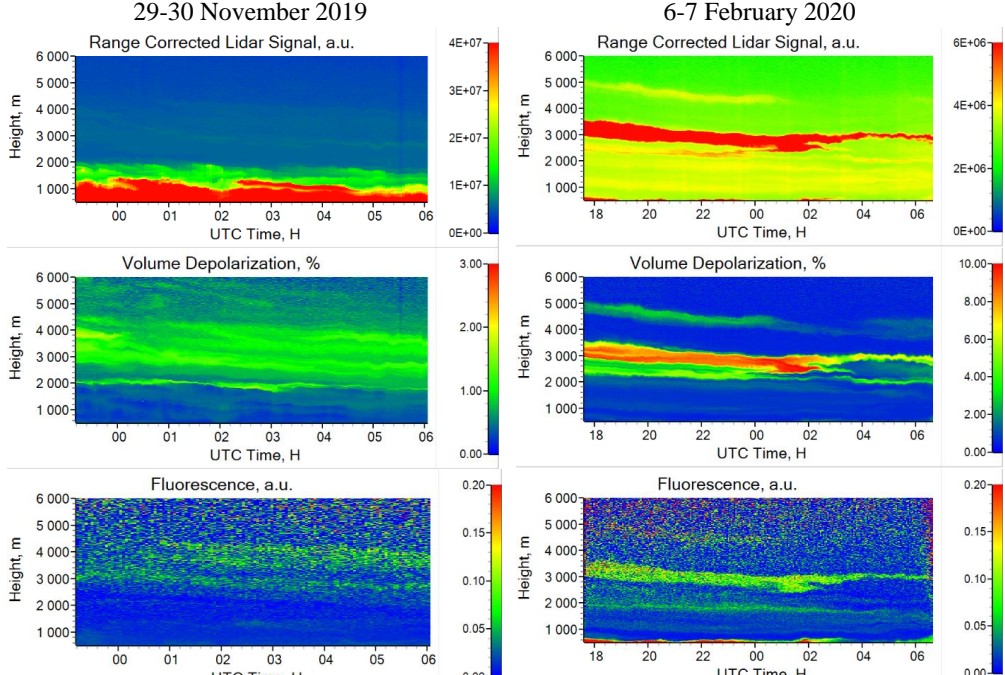

Fig.2. The range corrected lidar signal at 1064 nm, volume depolarization ratio $\delta_{1064}$ and
fluorescence backscattering measured at Lille, on 29-30 November 2019 (on the left) and 6-7
February 2020 (on the right).






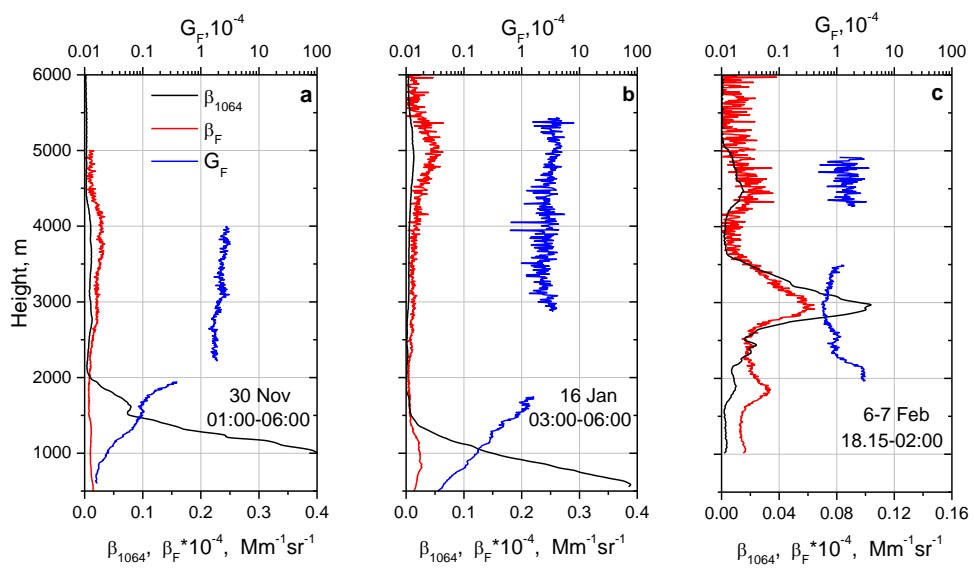

Fig.3 Vertical profiles of aerosol ($\beta_{1064}$) and fluorescence ($\beta_F$) backscattering coefficients
together with the fluorescence capacity ($G_F$) on (a) 30 November 2019, (b) 16 January 2020 and
(c) 6-7 February 2020.









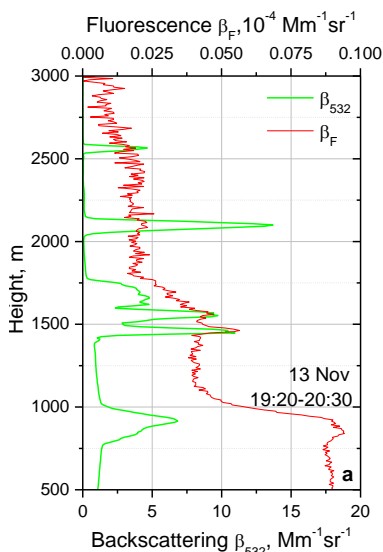 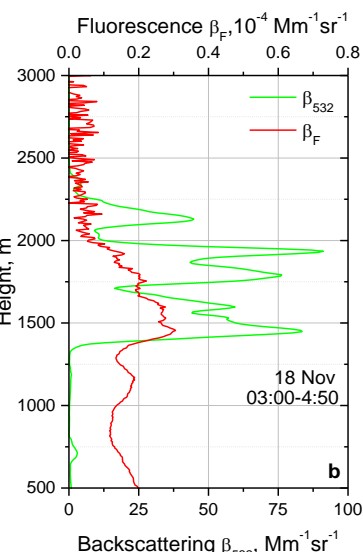

Fig.4. Aerosol ($\beta_{532}$) and fluorescence ($\beta_F$) backscattering coefficients on 13 and 18 November 2019.



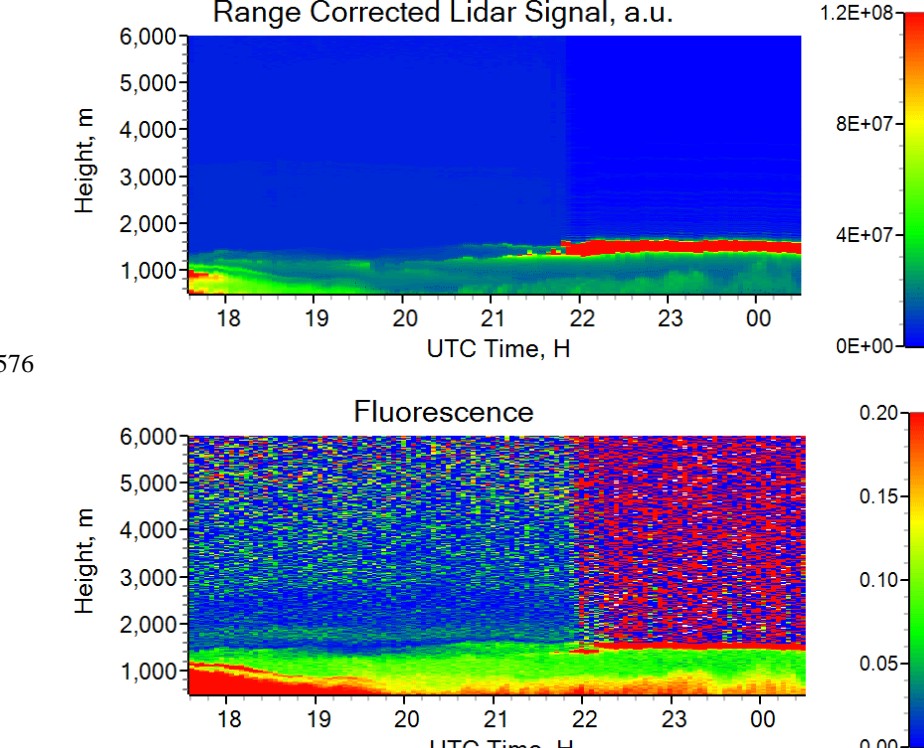




Fig.5. Height-temporal distribution of the range corrected lidar signal at 1064 nm and
fluorescence backscattering coefficient $\beta_F$ (in arbitrary units) on 19-20 November 2019.





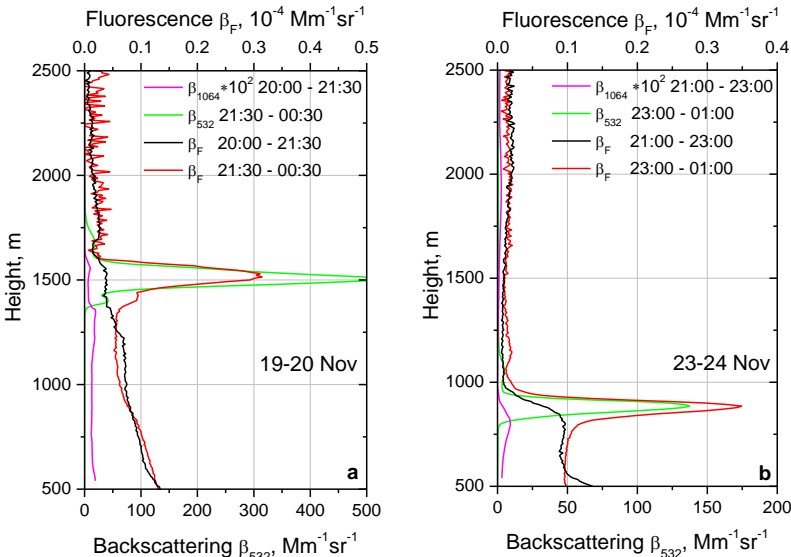

Fig.6. Aerosol and fluorescence backscattering coefficients on (a) 19-20 and (b) 23-24 November 2019 for two time intervals: prior and after cloud formation. Backscattering coefficient $\beta_{1064}$ prior to cloud formation is low, so it is multiplied by factor 100 to be distinguished at this figure.






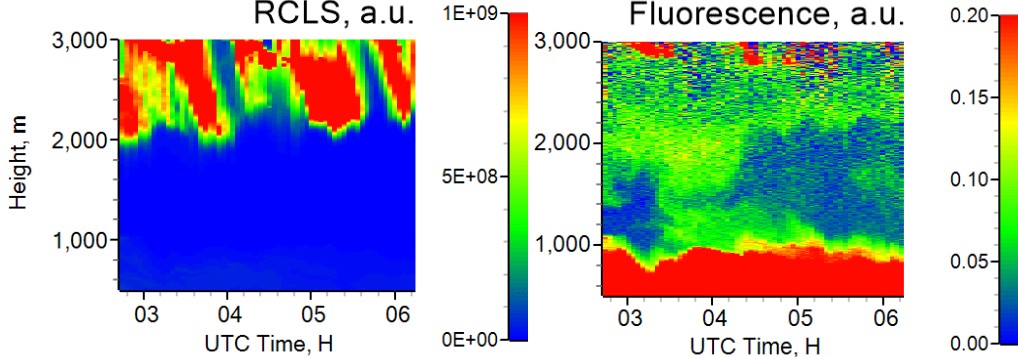


Fig.7. Height-temporal distribution of the range corrected lidar signal (RCLS) at 1064 nm and
the fluorescence backscattering coefficient on 15 November 2019.




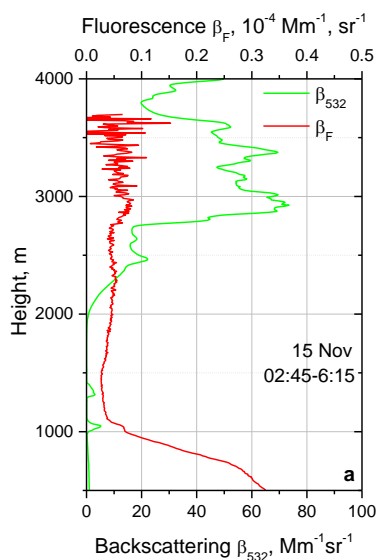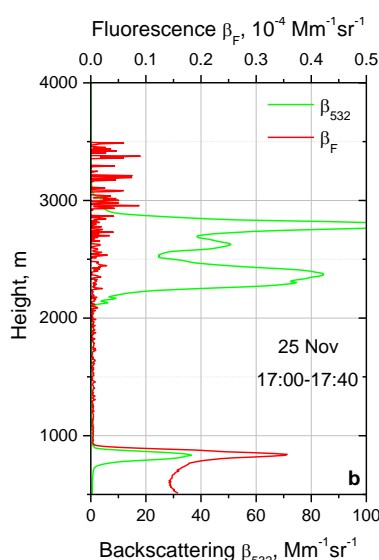

Fig.8. Aerosol ($\beta_{532}$) and fluorescence ($\beta_F$) backscattering coefficients on (a) 15 and (b) 25
November 2019.