# Peer review of "Combined use of Mie-Raman and fluorescence lidar observations for improving aerosol"

_Atmospheric Measurement Techniques, 2020_

## Referee Comment (RC1) · Anonymous Referee #1 · 14 Sep 2020

The authors show the results of an experiment dealing with Raman Scattering and fluorescence aiming studying aerosol properties. Innthis study a set of different scenarios are shown aiming to verify how meteorlogical conditions might affect the overall results. Also the presence of clouds were inspected. The experimental setup is well explained as well the theory involved in making this experiment. Given all this aspects I consider the paper is well suited for thi journal and reaches all the needed publication standards.

---

## Referee Comment (RC2) · Anonymous Referee #2 · 14 Sep 2020

General: The paper contains new and very interesting observations obtained with a new approach of a fluorescence lidar for aerosol characterization. This lidar feasibility study is clearly worthwhile to be published in AMT. Nevertheless, the interpretation of the observations needs to be improved. A clear and more systematic separation of the different fluorescence contributions would be helpful to better follow the discussion. The argumentation is partly week and a bit speculative.

Minor revisions are requested.

The abstract has to be updated and adjusted .... after all the suggested improvements.

P2, L39: Burton et al. 2012... Only one reference here? What about own papers:

Veselovskii et al., 2015, 2020, what about Tesche 2011, Tellus, SAMUM 2, what about all the efforts within the ACTRIS EARLINET group on aerosol typing... during the last five years.

P2, L44-47: I have my doubts that aerosol particles can be clearly identified and quantified in cloud layers. Ok, you can detect them, but it is already well known that interstitial aerosol particles are always present in clouds. It is impossible to have aerosol particle free clouds.

P3, L76-84: These are confusing statement. I am puzzled by the wording ..'external' vs 'internal' mixing of aerosol particles within liquid water droplets. There is only ONE scenario: It is impossible to have droplets without a CCN, and it is also impossible to have clouds without interstitial aerosol particles (non-activated particles). So, there is only this ONE scenario: a mixture of interstitial aerosol particles (not acting as CCN) and droplets, each of the droplets nucleated on a CCN. The CCNs may be completely dissolved in the droplet, or survived as a solid particle within the droplet, as is the case for dust or soot CCN. The interstitial aerosol particles (in the cloud) may be much larger than the particles outside the cloud (because of strong water uptake at 100% rel. humidity), so the aerosol backscatter efficiency of particles within the cloud may be larger by a factor of 5 and even more, compared to the aerosol backscatter outside the cloud layer.

I recommend to avoid to introduce ... internal and external mixtures! There is only this ONE scenario: interstitial aerosol particles and cloud droplets. Now we need a clear differentiation: What is the contribution of dry particles to fluorescence? What is the contribution of fully deliquescent (dissolved, solution) aerosol particles? Sulfate particles are fully dissolved at high humidities? Can we be sure that the fluorescence signal in clouds is exclusively from interstitial aerosol particles? No contribution by cloud droplets? That needs to be carefully discussed.

Dust particles with liquid shell .... produce an enhanced fluorescence signal (lens

effect)! Is that checked? Is there a reference for that?

P5, L157: Please keep in mind that RH increases from dry conditions (e.g., RH of 40%) to moist conditions (e.g., 80%, 90%, 95%) already a few hundred meters below cloud base, and then to 100% above cloud base. The aerosol particles grow by water uptake, change their backscatter efficiency and the fluorescence capability, some get liquid, some remain dry. Then in the cloud, cloud droplets come into play, backscatter efficiency of interstitial aerosol particles (at 100% humidity) may be much larger than for the aerosol particles below the cloud, or before the cloud formed. All this needs to be considered... in the cloud observations of backscatter and fluorescence signals.

Is all this known well enough to quantify the aerosol-related (non-droplet) fluorescence signal in clouds? To my opinion this is not the case. We do not know much about this. So, I have my doubts about Eq.(7).

P6, L176, the particle fluorescence capacity is introduced. I would recommend to do that in form of an equation. Now, the fluorescence signal will change with increasing humidity and water uptake and this in a different way as the total backscatter signal. Again, I think the knowledge about water uptake and the link to fluorescence signal changes is just qualitative.... So there is no clear knowledge about the increase of the capacity G with water uptake...

P6, L176, beta\_L ^a is introduced in Eq. (1) but not beta\_L or beta^a.

P6, L176: The capacity G\_F expresses what? The fluorescence signal changes probably when aerosol particle get a wet coating, the backscatter signal changes by water uptake, so there is no clear reference (denominator), and thus, what does G\_F indicate?

P6, L185 and L187: Again, because of the not well-known impact of water uptake, I do not belive that you can quantify N and V at conditions with rising humidity just below cloud base or even within the cloud? ... so that you can not estimate fluorescence

cross sections accurately enough. If you want to present it please clearly state that there are many questions how trustworthy this estimation is.

3. Observations....

A general comment: Trajectory analysis would be helpful for all cases discussed. There is no need to show them all, but it would improve the discussion .... to know more about the origin of air masses, and the kind of aerosol mixtures...

And it would also be helpful to have something like a bullet point list or an overview table, what aerosol produces fluorescence, what does not cause fluorescence, the same for droplets or water in aerosol particles..., what is producing fluorescence, and what does not.... And please provide references that support these statements

I ask for such a table because I learned more and more about fluorescence in detail .... from page to page of the manuscript, without having a complete picture right in the beginning of the discussion. Such an overview would facilitate all discussions and complex interpretation of the shown observations.

P7, L205-206: Do you mean external mixing of dust and biological /organic particles, or do you mean internal mixing, this would mean coating or partly coating of dust particles with organic material.

P7, L215: Pure water is not fluorescing, but what about the aqueous solution of dissolved aerosol particles (before becoming a droplet when acting as CCN). Again, what about the change in fluorescence efficiency with increasing water uptake and finally even change of phase (from dry and solid to totally liquid-acid aerosol particle)?

P8, in general and to mention again: would be nice to have HYSPLIT backward trajectories... to learn more about aerosol mixtures observed and what kind of aerosol are candidates for causing significant fluorescence.

P9, L266-280: Again the discussion part with N and V, I am not convinced that this is a fruitful part. Yes, there are numbers, but can we trust them?

3.2. Fluorescence of aerosol particles within clouds

This section is very interesting but, at the same time, a bit confusing. A more systematic way of presentation would be useful: What causes fluorescence, what not, what is the impact of water uptake, what happens with fluorescene when droplets are formed, with 'liquid' CCN and with solid CCN, etc.... So, a bullet point list or a Table... would be nice.

P10, L290-299: Again, this separation of externally and internally mixed cloud.... As mentioned above, there is only ONE scenario:

In all clouds, there is just a mixture of interstitial aerosol particles (not acting as CCN) and cloud droplets nucleated on the available CCNs.

Furthermore: In the cloud we have 100% relative humidity, so the interstitial aerosol particles are not dry, and most of them are just solution droplets (before activation .... to become cloud droplets). And the water droplets, on the other hand side, are formed on part of the solution droplets (CCN) but now consist almost entirely of water.

So please rephrase, and avoid external and internal mixing...

P10, L311: What do you mean here....? Fluorescent aerosol particles are inside the water particles. Ok, but must they be solid? If they are dissolved in the aqueous solutions, will there still be a fluorescence signal? May be droplets and CCN in the droplet do not produce any fluorescence signal, and fluorescence is only caused by the interstitial aerosol particles, and the increase in the fluorescence signal arises from water uptake effects?

So, what shows Fig4?  $\ldots$  in contrast to Fig5? If there is a difference, what is the reason?

P11, L322: Please, do not switch from one wavelength to another. That makes comparisons confusing. If beta1064 is 0.07 Mm-1 sr-1, then the 532 nm backscatter coefficients below the cloud is probably about 0.15 to 0.2 Mm-1 sr-1. That should be mentioned. And then we have an increase by a factor of roughly 3000 when you measure cloud a cloud beta532 value of 500 Mm-1 sr-1, and the fluorescence signal increases just by a factor of 5.... that is a good proof that water does not produce a fluorescence contribution. Please state that, if my comment is true, and if there is definitely no cross talk...

P11, L327-328: Again almost the same increase in beta 532 when we start from about 0.04 Mm-1 sr-1 for 532nm (estimated from 1064 nm information) and end up at 130 Mm-1 sr-1. All this should be given in more clearly way ... by using ONE wavelength.

But can we trust an increase by factor 5 of the aerosol-related fluorescence backscatter when the elastic backscatter increase by a factor of 3000? Can we be sure that there is no cross talk, not stray light somewhere, nothing? What causes the increase...? The lens effect? Is there another explanation?

P11, L342 For insoluble particles.... increase of fluorescence ... by lens effects.... Yes that can be, but it remains speculations, most of urban aerosol is sulfate aerosol (and not BC-containing haze) and sulfate particles dissolve completely ... and then there is no lens effect...

P12, L358-362: This is again a non-acceptable speculation. The two cloud layers may have formed in two different air masses with different aerosol types, and the different aerosol types caused different levels of fluorescence.

Figure 2, would be nice to have backward trajectories... and thus origin of air masses for the two cases.

Figure 3, again: what is the origin of the aerosol (according to HYSPLIT trajectories)?

Figure 4, strong increase of cloud beta532 but not of fluorescence beta, what can we conclude? Strong increase of beta532 by droplet backscatter, and at least significant increase of the fluorescence signal because of water uptake of interstitial particles ...

And again, HYSPLIT trajectories would be nice to all the cases discussed. The trajectories must not be shown in detail, but information about origin and mixture of aerosol particles would be helpful.

Final remark:

This is a good paper and needs only some minor clarifying information and a clear definition of the cloud environment (with interstitial non-CCN aerosol particles and CCNbased cloud droplets).

Afterwards (in the comparison...before vs within the cloud) a clear definition and systematic separation of the contributions by dry aerosol particles, wet aerosol particles, dissolved aerosol particles, cloud droplets nucleated on sulfate aerosol, and cloud droplets formed on soot and dust particles to fluorescence and to backscatter would be good and would make the discussion easier.

---

## Author Comment (AC1) · 1 Oct 2020

Response to Referee 1

The authors show the results of an experiment dealing with Raman Scattering and fluorescence aiming studying aerosol properties. Innthis study a set of different scenarios are shown aiming to verify how meteorlogical conditions might affect the overall results. Also the presence of clouds were inspected. The experimental setup is well explained as well the theory involved in making this experiment. Given all this aspects I consider the paper is well suited for thi journal and reaches all the needed publication standards.

[Figure]

We are grateful to the reviewer, for reading the manuscript and for appraisal of our work.

---

## Author Comment (AC2) · 1 Oct 2020

Response to Referee 2

First of all, we would like to thank the reviewer for careful reading the manuscript and for numerous useful suggestions. In revised manuscript, we changed the structure, adding more information about fluorescence in Introduction. We also strongly revised the chapter, containing analysis of particle fluorescence in the cloud.

Below we provide detailed response to the reviewer comments

*General: The paper contains new and very interesting observations obtained with a new approach of a fluorescence lidar for aerosol characterization. This lidar feasibility study is clearly worthwhile to be published in AMT. Nevertheless, the interpretation of the observations needs to be improved. A clear and more systematic separation of the different fluorescence contributions would be helpful to better follow the discussion. The argumentation is partly week and a bit speculative.*
*Minor revisions are requested.*

*The abstract has to be updated and adjusted: after all the suggested improvements.*

Abstract is revised and shortened

*P2, L39: Burton et al. 2012. Only one reference here? What about own papers: Veselovskii et al., 2015, 2020, what about Tesche 2011, Tellus, SAMUM 2, what about all the efforts within the ACTRIS EARLINET group on aerosol typing during the last five years.*

In revised manuscript we added:
"e.g. Tesche et al., 2011; Burton et al., 2012, Luís Guerrero-Rascado et al., 2018; Veselovskii et al., 2020 and references therein"
Definitely, there are a lot of important papers, showing potential of lidar technique for aerosol studies. We just can't mention them all, because we focus on the fluorescence measurements.

*P2, L44-47: I have my doubts that aerosol particles can be clearly identified and quantified in cloud layers. Ok, you can detect them, but it is already well known that interstitial aerosol particles are always present in clouds. It is impossible to have aerosol particle free clouds.*
*P3, L76-84: These are confusing statement. I am puzzled by the wording ..'external' vs 'internal' mixing of aerosol particles within liquid water droplets. There is only ONE scenario: It is impossible to have droplets without a CCN, and it is also impossible to have clouds without interstitial aerosol particles (non-activated particles). So, there is only this ONE scenario: a mixture of interstitial aerosol particles (not acting as CCN) and droplets, each of the droplets nucleated on a CCN. The CCNs may be completely dissolved in the droplet, or survived as a solid particle within the droplet, as is the case for dust or soot CCN.*
*The interstitial aerosol particles (in the cloud) may be much larger than the particles outside the cloud (because of strong water uptake at 100% rel. humidity), so the aerosol backscatter efficiency of particles within the cloud may be larger by a factor of 5 and even more, compared to the aerosol backscatter outside the cloud layer.*
*I recommend to avoid to introduce : internal and external mixtures! There is only this ONE scenario: interstitial aerosol particles and cloud droplets. Now we need a clear differentiation: What is the contribution of dry particles to fluorescence? What is the contribution of fully deliquescent (dissolved, solution) aerosol particles? Sulfate*

*particles are fully dissolved at high humidities? Can we be sure that the fluorescence signal in clouds is exclusively from interstitial aerosol particles? No contribution by cloud droplets? That needs to be carefully discussed.*

This is important comment and we agree with reviewer. In revised manuscript we don't use "internal or external mixture". Numerous modifications are introduced in the text. In particular, in Introduction we added paragraph:

"Interpretation of fluorescent measurements in a cloud is even more challenging. The liquid cloud is a mixture of interstitial aerosol particles (non-activated particles) and droplets, formed on the cloud condensation nucleus (CCNs). The CCNs may be completely dissolved in the droplet, or survived as a solid particle within the droplet, as is the case for dust or soot. The relative contributions of interstitial aerosol and activated CCNs in the droplets to the total cloud fluorescence backscatter are unknown, and the need to estimate these contributions was one of the motivations of this study"

In revised manuscript we strongly modified the section, describing fluorescence measurements in the clouds and Conclusion as well. The discussions that looked too speculative are removed.

*Dust particles with liquid shell produce an enhanced fluorescence signal (lens effect)! Is that checked? Is there a reference for that?*

To our knowledge, nobody discussed lens effect in respect to fluorescence. But physically, it may take place.

*P5, L157: Please keep in mind that RH increases from dry conditions (e.g., RH of 40%) to moist conditions (e.g., 80%, 90%, 95%) already a few hundred meters below cloud base, and then to 100% above cloud base. The aerosol particles grow by water uptake, change their backscatter efficiency and the fluorescence capability, some get liquid, some remain dry.*

Yes, influence of RH on fluorescence is an important question. RH strongly influences elastic scattering, but fluorescence is altered much less. We added paragraph in Introduction:

"One of the factors that intricate obtaining the quantitative information about aerosol properties from fluorescence measurements, is influence of the relative humidity (RH). The aerosol particles grow by water uptake, changing their elastic scatter cross-section, but the change in water percentage within an aerosol particle, normally does not alter the chemical components, so total amount of fluorescent molecules within a particle does not change. However the illumination intensity distribution within a particle, as well as the emission angle distribution will be altered by the change of particle size, shape and refractive index, and this modification may affect the fluorescence measurement. The phase functions of the microspheres for the incoherent scattering (fluorescence is an example of incoherent scattering), were computed in works of Kerker and Druger (1979) and Veselovskii et al. (2002a). Results demonstrate, that fluorescence of particles dissolved in water microspheres can be increased in the backward direction by factor ~2, comparing to fluorescence of a bulk material (calculated per gram of solid matter). This enhancement, however, occurs for relatively big microspheres with size parameter $x = \dfrac{2\pi r}{\lambda}$ exceeding approximately 10 (Veselovskii et al., 2002a). For the wavelength λ=532 nm corresponding radius *r* is about 1.0 μm, so fluorescence of the fine mode particles should be affected less by the hygroscopic growth. We should mention also that for insoluble particles, the presence of the water shell, at the condition of high RH, in principle, can lead to an additional increase of the fluorescence, due to the water droplet lens effect. Similar effect is well known for the soot particles covered by non-absorbing shell (Schnaiter, 2005)."

*Then in the cloud, cloud droplets come into play, backscatter efficiency of interstitial aerosol particles (at 100% humidity) may be much larger than for the aerosol particles below the cloud, or before the cloud formed. All this needs to be considered:in the cloud observations of backscatter and fluorescence signals.*

As we mentioned, fluorescence is less altered by droplets formation than elastic scattering. Estimated (from numerical modeling) increase of fluorescence backscattering should be about factor 2. Fig. 8a demonstrates that fluorescence of aerosol in the cloud increases about twice, when elastic backscattering is increased above 3 orders.

*Is all this known well enough to quantify the aerosol-related (non-droplet) fluorescence signal in clouds? To my opinion this is not the case. We do not know much about this. So, I have my doubts about Eq.(7).*

Definitely, there are a lot of questions, when we try characterize aerosol inside the cloud from fluorescence. Still we think, that for the aerosol outside of the cloud such estimations can be done, at least at the condition of low RH. Corresponding comment is added to the manuscript.

*P6, L176, the particle fluorescence capacity is introduced. I would recommend to do that in form of an equation.*

Done

*Now, the fluorescence signal will change with increasing humidity and water uptake and this in a different way as the total backscatter signal.*
*Again, I think the knowledge about water uptake and the link to fluorescence signal changes is just qualitative. So there is no clear knowledge about the increase of the capacity G with water uptake:*

We agree, that altering the fluorescence by water uptake is not completely understood and results, at least for the clouds, are qualitative. Corresponding comments are added to the text..

*P6, L176, beta_L ˆa is introduced in Eq. (1) but not beta_L or betaˆa.*

Yes, but on page 7 we write "Here and below, for simplicity, we will use notation $\beta^a \equiv \beta$ ." So we think here is no misunderstanding.

*P6, L176: The capacity G_F expresses what? The fluorescence signal changes probably when aerosol particle get a wet coating, the backscatter signal changes by water uptake, so there is no clear reference (denominator), and thus, what does G_F indicate?*

For the fine mode particles water uptake should not alter significantly fluorescence backscattering. For example, on 29-30 November variation of RH from 70% to 20% is accompanied by decrease of elastic scattering by factor 40 while fluorescence backscattering is decreased less than twice. So drop of fluorescence capacity is directly related with water uptake and this is important parameter when we compare different aerosols at low RH. Besides, at low RH the fluorescence capacity is used to compare fluorescent properties of different aerosol types.

*P6, L185 and L187: Again, because of the not well-known impact of water uptake, I do not believe that you can quantify N and V at conditions with rising humidity just below cloud base or even within the cloud? : So that you can not estimate fluorescence cross sections accurately*

*enough. If you want to present it please clearly state that there are many questions how trustworthy this estimation is.*

In the revised manuscript we describe the challenges of interpretation the fluorescence measurements. Estimations of the fluorescence cross section are presented only for low RH, when water uptake does not alter the results.

*3. Observations: : :.*
*A general comment: Trajectory analysis would be helpful for all cases discussed. There is no need to show them all, but it would improve the discussion .... to know more about the origin of air masses, and the kind of aerosol mixtures*

Trajectory analysis is added.

*And it would also be helpful to have something like a bullet point list or an overview table, what aerosol produces fluorescence, what does not cause fluorescence, the same for droplets or water in aerosol particles, what is producing fluorescence, and what does not. And please provide references that support these statements*
*I ask for such a table because I learned more and more about fluorescence in detail :from page to page of the manuscript, without having a complete picture right in the beginning of the discussion. Such an overview would facilitate all discussions and complex interpretation of the shown observations.*

In revised manuscript we moved the information about fluorescence measurements to the Introduction, which should facilitate the process of manuscript reading. We wouldn't like to provide the table, because the information about fluorescence of different atmospheric aerosols in ambient conditions is quite rare. For the same reason it is not easy to support our conclusions with references: there are quite few publications on fluorescence lidars, providing quantitative results.

*P7, L205-206: Do you mean external mixing of dust and biological /organic particles, or do you mean internal mixing, this would mean coating or partly coating of dust particles with organic material.*

It can be both. Unfortunately, at this stage we can not separate these two possible scenarios.

*P7, L215: Pure water is not fluorescing, but what about the aqueous solution of dissolved aerosol particles (before becoming a droplet when acting as CCN). Again, what about the change in fluorescence efficiency with increasing water uptake and finally even change of phase (from dry and solid to totally liquid-acid aerosol particle)?*

Dissolved aerosol particles should provide the fluorescence. From our expectations, fluorescence efficiency in dissolved state should be increased by approximately factor 2. However, in some measurements, when cloud was formed at the top of aerosol layer, this enhancement was up to factor 5. At a moment we can not identify the mechanisms, responsible for such strong enhancement.

*P8, in general and to mention again: would be nice to have HYSPLIT backward trajectories to learn more about aerosol mixtures observed and what kind of aerosol are candidates for causing significant fluorescence.*

HYSPLIT analysis is added

*P9, L266-280: Again the discussion part with N and V, I am not convinced that this is a fruitful part. Yes, there are numbers, but can we trust them?*

We think, that such estimations can be used, at least for low RH. We compared results of such estimations with regularization inversion and agreement is reasonably good.
*3.2. Fluorescence of aerosol particles within clouds*
*This section is very interesting but, at the same time, a bit confusing. A more systematic way of presentation would be useful: What causes fluorescence, what not, what is the impact of water uptake, what happens with fluorescene when droplets are formed, with 'liquid' CCN and with solid CCN, etc. So, a bullet point list or a Table would be nice.* We added a section in Introduction, trying to explain these issues.

The section is strongly modified and some of unsupported statements are removed.

*P10, L290-299: Again, this separation of externally and internally mixed cloud: As mentioned above, there is only ONE scenario: In all clouds, there is just a mixture of interstitial aerosol particles (not acting as CCN) and cloud droplets nucleated on the available CCNs. Furthermore: In the cloud we have 100% relative humidity, so the interstitial aerosol articles are not dry, and most of them are just solution droplets (before activation to become cloud droplets). And the water droplets, on the other hand side, are formed on part of the solution droplets (CCN) but now consist almost entirely of water. So please rephrase, and avoid external and internal mixing*

We agree with reviewer. "Internal or external mixing" is not used in the revised manuscript.

*P10, L311: What do you mean here? Fluorescent aerosol particles are inside the water particles. Ok, but must they be solid? If they are dissolved in the aqueous solutions, will there still be a fluorescene signal? May be droplets and CCN in the droplet do not produce any fluorescence signal, and fluorescence is only caused by the interstitial aerosol particles, and the increase in the fluorescence signal arises from water uptake effects?*

We think that dissolved CCN provides the fluorescence signal. As mentioned, in dissolved state the signal should be about factor 2 stronger than in solid. Still this signal is probably rather week. In particular, Fig.8b demonstrates, that fluorescence of the cloud at 2500 m is low, when the aerosol below the cloud base is not detectable. This is in contrast with results in Fig.8a, where strong fluorescence is probably provided by interstitial aerosol

*So, what shows Fig4? in contrast to Fig5? If there is a difference, what is the reason?*

In Fig.4 oscillations of elastic backscattering don't lead to synchronous oscillations of fluorescence, while in Fig.5 they do. We have no ultimate explanation for such difference, but looks like in Fig.5 the vapor is condensed in the aerosol layer, while in Fig.4 the weak water cloud layers are just mixed with aerosol.

*P11, L322: Please, do not switch from one wavelength to another. That makes comparisons confusing. If beta1064 is 0.07 Mm-1 sr-1, then the 532 nm backscatter coefficients below the cloud is probably about 0.15 to 0.2 Mm-1 sr-1. That should be mentioned. And then we have an increase by a factor of roughly 3000 when you measure cloud a cloud beta532 value of 500 Mm-1 sr-1, and the fluorescence signal increases just by a factor of 5. that is a good proof that water does not produce a fluorescence*

*contribution. Please state that, if my comment is true, and if there is definitely no cross talk: : :*

In revised manuscript we provide backscattering at 532 nm prior and after cloud formation. We believe that there is no cross talk, because we had observations (e.g. Fig.8b) with strong cloud backscattering, which were not accompanied by increase of fluorescence.

*P11, L327-328: Again almost the same increase in beta 532 when we start from about 0.04 Mm-1 sr-1 for 532nm (estimated from 1064 nm information) and end up at 130 Mm-1 sr-1. All this should be given in more clearly way : : : by using ONE wavelength.*

Done

*But can we trust an increase by factor 5 of the aerosol-related fluorescence backscatter when the elastic backscatter increase by a factor of 3000? Can we be sure that there is no cross talk, not stray light somewhere, nothing?*

We believe that there is no cross talk, because we had observations (e.g. Fig.8b) with strong cloud backscattering, which were not accompanied by increase of fluorescence.

*What causes the increase: : :.? The lens effect? Is there another explanation?*

At a moment we can not identify the mechanism

*P11, L342 For insoluble particles increase of fluorescence by lens effects. Yes that can be, but it remains speculations, most of urban aerosol is sulfate aerosol (and not BC-containing haze) and sulfate particles dissolve completely and then there is no lens effect*

Yes, we can not prove it and we just mention such possibility. To our knowledge nobody considered lens effect in respect to fluorescence. But physically this is possible, for example for soot or dust particles covered by water shell. And definitely no lens effect for dissolved aerosol.

*P12, L358-362: This is again a non-acceptable speculation. The two cloud layers may have formed in two different air masses with different aerosol types, and the different aerosol types caused different levels of fluorescence.*

Yes, we agree that two cloud layers can be different. We just wanted to emphasize, that in some cases aerosol fluorescence in the cloud is very low (Fig.8b), while in others fluorescence is strong (Fig.8a), meaning significant content of aerosol in the cloud.

*Figure 2, would be nice to have backward trajectories: : : and thus origin of air masses for the two cases.*

We added information about back trajectories in the text.

*Figure 3, again: what is the origin of the aerosol (according to HYSPLIT trajectories)?*

HYSPLIT analysis is added

*Figure 4, strong increase of cloud beta532 but not of fluorescence beta, what can we conclude? Strong increase of beta532 by droplet backscatter, and at least significant increase of the fluorescence signal because of water uptake of interstitial particles*

We think that cloud layers, characterized by small content of aerosol (and so by low fluorescence), are mixed with aerosol particles. High RH in these layers doesn't alter significantly the fluorescence efficiency of aerosol, so oscillations of elastic backscattering is not accompanied by synchronous oscillation of fluorescence.

*And again, HYSPLIT trajectories would be nice to all the cases discussed. The trajectories must not be shown in detail, but information about origin and mixture of aerosol particles would be helpful.*

We added HYSPLIT analysis in the text for cases with clouds and elevated aerosol layers.

*Final remark:*
*This is a good paper and needs only some minor clarifying information and a clear definition of the cloud environment (with interstitial non-CCN aerosol particles and CCNbased cloud droplets). Afterwards (in the comparison: : :before vs within the cloud ) a clear definition and systematic separation of the contributions by dry aerosol particles, wet aerosol particles, dissolved aerosol particles, cloud droplets nucleated on sulfate aerosol, and cloud droplets formed on soot and dust particles to fluorescence and to backscatter would be good and would make the discussion easier.*

We tried to follow these suggestions in the revised manuscript.

---

## Referee Comment (RC3) · Anonymous Referee #3 · 12 Oct 2020

Summary: The manuscript reports on fluorescence measurements of atmospheric aerosols with a multi-wavelength Raman lidar, where the interference filter in the water vapor Raman channel was replaced by a broadband filter around 466 nm.

Although the study contains some interesting approaches, e.g. the possible synergy of combined measurements with multi-wavelength Raman lidar and fluorescence lidar, it is incomplete and too speculative at this stage and requires substantial extensions and improvements for a possible publication. For example, it is incomprehensible why the authors do not present aerosol events that could show the real strength of their modified lidar system (microphysical retrieval plus fluorescence), but only those that are actually

not suitable. A little more patience would have been necessary here. Furthermore, the paper shows technical weaknesses in both the experiment and the analysis, and the interpretation of the measurements is highly speculative. For example, the fluorescence measurement has not been thoroughly calibrated, and no backward trajectories were used for aerosol typing. Furthermore, the measurement results are discussed using relative humidity, although neither water vapor measurements with the lidar nor local radiosondes were available. Interestingly, the authors themselves point out some of these weaknesses in their conclusions, they should fix them and then resubmit the manuscript.

Major issues:

1. The calibration of the lidar was not performed with a spectral lamp (l. 169 ff), so the measurement trueness is questionable, and the authors are aware of this (l. 380 ff). Why was the calibration not performed? Nevertheless, the measurements are quantitatively evaluated and interpreted, this is not a consistent approach.

2. The authors speculate about the presence of aerosol mixtures (l. 204 ff). This can only be investigated with spectrometric fluorescence lidars, if at all. But at least an analysis of the backward trajectories should have been performed. This also applies to the statements regarding the change of G_F (l. 225 ff).

3. Relative humidity is used for the interpretation of the measurements, although it is not known sufficiently for these purposes, especially for hygroscopic aerosol growth (l. 211 ff). Thus the interpretation is a speculation.

4. Particle depolarization is not only a function of particle shape but also of particle size, this should be considered in the discussion.

5. The whole microphysical interpretation (l. 244 ff) is pure speculation. Why did the authors not wait for aerosol measurement cases where they could have used the strengths of their multi-wavelength Raman lidar?

[Figure]

6. The reviewer is sceptical about the measurements in chapter 3.2, which are supposed to prove an internal mixture of aerosol particles and cloud droplets (l. 311 ff). It is noticeable that the fluorescence signal associated with the cloud layers seems to be a function of the measurement height: below 1000 m very high 'fluorescence' values are found in clouds, around 1500 m slight increases, and above 1700 m elastic and fluorescence signals are uncorrelated. This may (but of course does not have to) indicate instrumental effects (height-dependent angle-of-incidence distribution of the backscattered photons). Are there measurement examples where liquid water clouds below 1000 m do not show increased fluorescence?

Minor issues:

1. The authors claim that lidars with spectrometers are less sensitive than those with standard detection channels (l. 61 ff). However, a comparison with published spectrometric measurements seems to contradict this. Please explain in more detail.

2. The authors plan to reduce the bandwidth of the interference filter for fluorescence measurements by a factor of 2 or even 4 in the future (l. 404 ff). However, this would further increase the measurement duration, which is already very long. Please explain in more detail.

Wording:

1. To speak of a 'highly efficient lidar operation' (l. 368) when in fact hour-long integration times are needed for fluorescence measurements is quite a stretch.

Type setting:

1. All variables in the running text and in the equations must be checked for correct math format. There are many formatting errors, for instance, variables are not italic (e.g., l. 133), or subscripts are italic (e.g., l. 119).

Figures:

[Figure]

1. Figures 1 and 7 are of poor quality.

2. Figure 6, colors for beta_1064 and beta_F are hardly distinguishable when printed.

---

## Author Comment (AC3) · 14 Oct 2020

First of all, we would like thank the Reviewer for careful reading our manuscript and suggestions. Below are responses to his comments.

Summary: The manuscript reports on fluorescence measurements of atmospheric aerosols with a multi-wavelength Raman lidar, where the interference filter in the water vapor Raman channel was replaced by a broadband filter around 466 nm. Although the study contains some interesting approaches, e.g. the possible synergy of combined measurements with multi-wavelength Raman lidar and fluorescence lidar, it is incomplete and too speculative at this stage and requires substantial extensions and improvements for a possible publication. For example, it is incomprehensible why the authors do not present aerosol events that could show the real strength of their modified lidar system (microphysical retrieval plus fluorescence), but only those that are actually not suitable. A little more patience would have been necessary here.

As mention by the Reviewer, combining the multiwavelength and fluorescence measurements can be a promising approach for aerosol characterization. But to implement it, some important questions should be answered first. These questions, in particular, are:

- Is fluorescence technique sensitive enough to be useful for lidar aerosol measurements, when part of the spectrum is selected by the interference filter?

- How the fluorescence signal is affected by the variation of the relative humidity and by the droplets formation?

- Is it possible to measure the fluorescence signal inside the cloud layer? In our manuscript we tried to get answers for these questions and to demonstrate the feasibility of our system for fluorescence studies.

By today we have measurements performed during high aerosol loading, and combining of multiwavelength retrievals with fluorescence data is in progress. However this is a subject of separate study. We wouldn't want to add multiwavelength inversion to this one.

We should recall also, that when aerosol near the cloud base is considered, usually the aerosol extinction coefficients are quite low and traditional multiwavelength Raman technique does not work. In our study we consider cases with low aerosol loading and suggest approach, based on use of predefined aerosol models, for aerosol characterization.

**Furthermore, the paper shows technical weaknesses in both the experiment and the analysis, and the interpretation of the measurements is highly speculative. For example, the fluorescence measurement has not been thoroughly calibrated,**

Equation (9) for the fluorescence backscattering contains the ratio of efficiencies of fluorescence and Raman channels. The dichroic optics used, allows efficient separation of fluorescence and Raman signals, so main source of uncertainty is relative sensitivity of PMTs in the channels. To equalize sensitivities, the PMT from fluorescence channel was installed in the Raman one and by small adjusting of voltage the same value of nitrogen Raman signal was obtained. The cathode sensitivity of R9880U-01 PMT between 387 nm and 466 nm changes for less than 15%, thus we assume that sensitivities of PMTs in both channels are the same and only difference in transmission of interference filters was considered. We estimate that uncertainty of such calibration is less than factor 2, which is sufficient for our purpose, because relative variations of

fluorescence backscattering coefficient are considered. Corresponding comment is added to the text.

and no backward trajectories were used for aerosol typing.

In revised manuscript we discuss backward trajectories

Furthermore, the measurement results are discussed using relative humidity, although neither water vapor measurements with the lidar nor local radiosondes were available.

Yes, RH data were available only from radiosond in Belgium (95 km away). However in this study we don't analyze the hygroscopic growth. RH data are taken as qualitative only.

Interestingly, the authors themselves point out some of these weaknesses in their conclusions, they should fix them and then resubmit the manuscript.

We definitely understand all these weak points, still we think that this study presents new important results.

**Major issues:**

1. The calibration of the lidar was not performed with a spectral lamp (l. 169 ff), so the measurement trueness is questionable, and the authors are aware of this (l. 380 ff). Why was the calibration not performed? Nevertheless, the measurements are quantitatively evaluated and interpreted, this is not a consistent approach.

We have already responded this comment. Discussing the cross sections obtained, we emphasize, that these are only rough estimations.

2. The authors speculate about the presence of aerosol mixtures (l. 204 ff). This can only be investigated with spectrometric fluorescence lidars, if at all. But at least an analysis of the backward trajectories should have been performed.

In revised manuscript we added back trajectory analysis. Air masses pass Africa and particle depolarization ratio is high. So dust is predominant in aerosol mixture.

This also applies to the statements regarding the change of GF (l. 225 ff).

We are not able indentify aerosol type for this case

3. Relative humidity is used for the interpretation of the measurements, although it is not known sufficiently for these purposes, especially for hygroscopic aerosol growth (l. 211 ff). Thus the interpretation is a speculation.

We don't analyze hygroscopic growth. We just say that RH is high at 1000 m and drops above 2000 m. The sonde data from England and from Belgium lead to similar conclusion.

4. Particle depolarization is not only a function of particle shape but also of particle size, this should be considered in the discussion.

Yes, depolarization definitely depends on particle size. But here we focus on the fluorescence. Analysis of dependence of depolarization on particle parameters is out of the scope of this study. 5. The whole microphysical interpretation (l. 244 ff) is pure speculation. Why did the authors not wait for aerosol measurement cases where they could have used the strengths of their multi-wavelength Raman lidar?

In situations, when aerosol extinction is low (for example when aerosol near the cloud base is analyzed) the multiwavelength Raman technique can not be used. So other approaches, allowing at least qualitative estimations of particle properties are needed. The estimations of particle properties, based on predefined aerosol models are widely used in remote sensing. In our study we used the aerosol models based on AERONET observations. Still we agree that such estimations need numerous assumptions, thus results obtained can be considered as qualitative only. Corresponding comments are added to revised manuscript.

**Discussion** paper**

6. The reviewer is sceptical about the measurements in chapter 3.2, which are supposed to prove an internal mixture of aerosol particles and cloud droplets (l. 311 ff).

**Reviewer 2 provided numerous comments, concerning "internal and external mixing". So in revised manuscript we don't use this terminology.**

It is noticeable that the fluorescence signal associated with the cloud layers seems to be a function of the measurement height: below 1000 m very high 'fluorescence' values are found in clouds, around 1500 m slight increases, and above 1700 m elastic and fluorescence signals are uncorrelated. This may (but of course does not have to) indicate instrumental effects (height-dependent angle-of-incidence distribution of the backscattered photons). Are there measurement examples where liquid water clouds below 1000 m do not show increased fluorescence?

Yes, reviewer is right, height dependence of fluorescence is complicated and depends on aerosol loading. We think that this is result of water uptake by aerosol (aerosol dissolving, water shell forming...). We don't see how instrumental effects can result in such profiles, because we had many aerosol observations without such "exotic" behavior at low altitudes (Fig.6 in this manuscript). Still in the presence of high RH elastic scattering and fluorescence don't correlate, because water uptake by particles normally does not increase fluorescence significantly.

**Minor issues:**

1. The authors claim that lidars with spectrometers are less sensitive than those with standard detection channels (l. 61 ff). However, a comparison with published spectrometric measurements seems to contradict this. Please explain in more detail.

Transmission of the interference filters used is above 95%, while transmission of grating spectrometer with fiber input is definitely lower. This why we say that spectrometer is less sensitive. In revised manuscript we modified this phrase as

"However, sensitivity of such lidar spectrometers is lower when compared to the technique based on selection of fluorescence spectrum intervals with interference filters, because the transmission of modern filters exceeds 90%".

2. The authors plan to reduce the bandwidth of the interference filter for fluorescence measurements by a factor of 2 or even 4 in the future (l. 404 ff). However, this would further increase the measurement duration, which is already very long. Please explain in more detail.

In Fig.2 we show fluorescence maps obtained with 2 min resolution at low aerosol loading. So we have resource to reduce the filter width. But our experience of fluorescence measurements (and data analysis) shows that we never have "too much" signal. So the phrase about bandwidth reduction is removed from revised manuscript.

Wording:

1. To speak of a 'highly efficient lidar operation' (l. 368) when in fact hour-long integration times are needed for fluorescence measurements is quite a stretch.

Changed for "efficient"

Type setting:

1. All variables in the running text and in the equations must be checked for correct math format. There are many formatting errors, for instance, variables are not italic (e.g., l. 133), or subscripts are italic (e.g., l. 119).

Changed

1. Figures 1 and 7 are of poor quality.

Why? We don't think that these are of poor quality...

2. Figure 6, colors for beta\_1064 and beta\_F are hardly distinguishable when printed.

We changed color of beta 1064 line for blue.